# Comparative Analysis of the Visual, Refractive and Aberrometric Outcome with the Use of 2 Intraocular Refractive Segment Multifocal Lenses

**DOI:** 10.3390/jcm13010239

**Published:** 2023-12-31

**Authors:** Bartłomiej Markuszewski, Adam Wylęgała, Nóra Szentmáry, Achim Langenbucher, Anna Markuszewska, Edward Wylęgała

**Affiliations:** 1Chair and Clinical Department of Ophthalmology, Faculty of Medical Sciences, Zabrze Medical University of Silesia, 40-760 Katowice, Poland; adam.wylegala@gmail.com (A.W.);; 2Wrocławskie Centrum Okulistyczne, 50-231 Wrocław, Poland; 3Dr. Rofl M. Schwiete Center for Limbal Stem Cell and Aniridia Research, Saarland University, 66424 Homburg, Germany; nszentmary@gmail.com; 4Department of Experimental Ophthalmology, Saarland University, 66424 Homburg, Germany; 5Department of Ophthalmology, District Railway Hospital, 40-760 Katowice, Poland

**Keywords:** aberrometer, multifocal IOL, ray tracing, spherical aberrations

## Abstract

To demonstrate the results of ray tracing higher- and lower-order aberrations in pseudophakic eyes with rotationally asymmetrical segment multifocal lenses, total high- and low-order aberrations, measured by root mean square value (RMS), refraction, uncorrected distance and uncorrected near visual acuity (UCDVA and UCNVA), and tear break-up time, were measured at scotopic size in 42 eyes of patients implanted with bifocal refractive Mplus15/Mplus30 IOL with +1.5 dpt near addition (42 eyes of patients implanted with Mplus15)/+3.0 dpt near addition (91 eyes of patients implanted with Mplus30), and 107 eyes of control group. No significant differences were noticed between the examined groups concerning UCDVA, UCNVA, and tear break-up time (*p* < 0.001). Coma and total high-order aberrations were significantly higher for the Mplus30 lens in comparison to the Mplus15 lens and the control group (Coma, Trefoil *p* < 0.001, Secondary Astigmatism *p* = 0.002). The spherical aberrations were significantly higher in the lower-addition lens (*p* = 0.016) in comparison to the control group and to the higher-addition lens group (*p* < 0.001). Both intraocular lens models were successful at reaching refractive aim, good distance, and near function with the lower higher-order aberrations for the low-addition lens.

## 1. Introduction

Cataract surgery or refractive lens exchange with the implantation of multifocal intraocular lens (MF IOL) allows for good visual performance and spectacle independence, particularly for the newer trifocal platforms (AT Lisa tri 839 MP, Panoptix, Fine Vision, among others), which lets the patients have good unaided near, intermediate, and far vision [1,2,3,4,5,6,7]. The aim of the surgery is to provide rapid and complete visual rehabilitation with marginal postoperative refractive errors, and sufficient uncorrected visual acuity at the three object distances. In cases of significant residual ametropia, additional photorefractive procedures may be needed [8]. An additional approach has been the use of rotational asymmetric intraocular lenses. These platforms exhibit refractive rotational asymmetry unlike the standard bifocal or trifocal intraocular lenses, which are rotationally symmetric and based on the principles of diffraction and refraction. Studies have reported excellent visual outcomes and low occurrence of photic phenomena following the use of these multifocal rotational asymmetric intraocular lenses. The amount of addition varies between lens models (usually +3.00 D for near vision, or +1.50 D for intermediate vision models) [9,10]. Aberrations refer to the differences in the optical path lengths, which represent the deviation between the actual and ideal wavefront achieved in the optical system. Optical aberrations for a specific visible light wavelength traveling through an optical system are generally classified into two broad categories. The first category includes lower-order aberrations, which account for most of the degradation in image quality. These lower-order aberrations comprise spherical refractive error (defocus) and cylindrical refractive error. The second category is higher-order aberrations (HOA), encompassing spherical aberration, coma, and others. When large HOA can impact the quality of vision and, notably, cannot be corrected with spectacles or contact lenses [11]. Light scattering can also, along with HOAs, further degrade retinal image quality [12]. Optical aberrations may differ depending on many features of the intraocular lenses, including the amount of near addition. Although, currently there is no consensus about the suitability of aberrometers to evaluate the clinical performance of multifocal intraocular lenses [13,14,15], it seems that they can provide useful information [16]. The present study aimed to compare visual, refractive, and aberrometric outcomes following surgery with the use of two refractive central-sparing rotational asymmetric multifocal lenses with two different amounts of near vision addition, specifically +1.5 versus +3.0 dpt.

## 2. Materials and Methods

### 2.1. Patients

This retrospective observational comparative study included patients following the implantation of rotationally asymmetrical lens models during cataract or refractive lens exchange surgery (either model Lentis LS313MF15, LS313MF15T or model Lentis LS313MF30, LS313MF30T). The control group consisted of patients who were admitted to hospital for treatment of refractive error by either excimer laser refractive surgery or refractive lens exchange surgery. The qualification examination for laser refractive procedures included corneal tomography, refraction analysis, and biometry. This study adhered to the tenets of the Declaration of Helsinki and was approved by the local bioethical committee (registration number 6/PNDR/2021). Informed consent was not required due to meeting the criteria that “the research involves no more than minimal risk to the subjects” [17].

The inclusion criteria for the surgery were moderate or high refractive error not qualified for the laser corneal correction without or with the coincidence of incipient or moderate cataract and presbyopia. The main exclusion criteria for excimer laser refractive procedure were the pachymetry analysis on the corneal tomography, showing too thin cornea below 500 um, irregular cornea, too steep keratometry of above 46 dpt, or too flat keratometry of below 39 dpt. The exclusion criteria for the study were active ocular disease, such as cornea dystrophy, maculopathy, glaucoma, and also ectatic disease, previous keratorefractive procedures and any intraocular surgical procedures, and illiteracy. Previously undergone keratorefractive procedures cause inaccurate measures of anterior keratometry and variation in keratometric index. Therefore, specific intraocular lens calculation methods are needed with different accuracy [18]. All patients underwent cataract surgery or refractive lens exchange between 2011 and 2021. Eligible patients for this study included patients between 27 and 81 years of age. All patients were admitted for surgery at the Wroclaw Ophthalmology Center, Wroclaw, Poland, as an outpatient procedure.

### 2.2. Group and Intraocular Lenses

In this study, patients received a refractive central-sparing rotational asymmetric multifocal lens with an addition of +1.5 dpt sph Lentis Mplus Comfort LS313MF15, LU313MF15T (Mplus15) Oculentis BV, Arnhem, The Netherlands (Teleon Surgical BV, AV Spankeren, The Netherlands), and a refractive central-sparing rotational asymmetric multifocal lens with the addition of +3.0 dpt sph Lentis LS313MF30, LS313MF30T (Mplus 30) (Teleon surgical BV) for routine clinical practice. Lentis Mplus15 and Mplus30 (Teleon surgical BV) distributed by Topcon Polska Sp. z o.o. are refractive, rotationally asymmetric lenses that contain an aspheric distance vision zone combined with a +3.00 dpt near vision zone located at the posterior surface. The Mplus15 differs in the near vision zone addition of +1.50 dpt, often classified today as an extended depth of focus (EDOF) lens. In theory, light passing through the transition sector is refracted away from the optical axis; hence, the image is imaged to the object space with a shorter focal distance. Superposition of interference or diffraction is avoided [19]. The biconvex lens with a 6.0 mm optic and 12.0 mm overall length is built of acrylic copolymer with ultraviolet filtering and a hydrophobic surface. The design is shown in Figure 1. 

The intraocular lens power calculation was based on the biometrics from IOL Master v.5.0 (Carl Zeiss, Jena, Germany) using the SRK-T formula and double check by Barrett Universal formula for all eyes with an axial length above 22.00 mm and Hoffer-Q formula with cross check by Barrett Universal formula for eyes below the axial length of 22.00 mm. The implant power for the previously myopic eyes was selected, targeting between emmetropia to −0.25 dpt sph postoperative refractive power. A constant of 118.5 for the SRK-T formula and Barrett Universal formula and a pACD constant of 5.21 was used for the Hoffer-Q formula. The implant power for previously hyperopic eyes was selected, targeting emmetropia to +0.25 dpt sph postoperative refractive power. The lens models were selected according to the individual needs of the patients (work, hobby, etc.).

### 2.3. Surgical Technique

All the patients underwent routine phacoemulsification surgery. The surgeries were performed by two experienced surgeons (Jolanta Markuszewska Żelbromska, and Bartlomiej Markuszewski). Revisions with replacement of the studied lens by monofocal lenses were not required due to neuroadaptive failure or any other reason in the studied groups. In all cases, anesthesia was accomplished by a retrobulbar block of preservative-free lidocaine 2.0% with mild sedation of intravenous midazolam (1 mg to 4 mg) performed by the surgeon, and pharmacological mydriasis was induced using 1.0% tropicamide and 0.1% phenylephrine. The IOL was implanted with the injector Viscojet 2.2 through a 2.2 mm main clear cornea incision. The IOL was aligned and positioned according to the horizontal marks made on the corneal limbus at 9.00 and 3.00 clock hours with a sterile marker at the slit lamp just prior to surgery in the vertical patient position. The patient’s head was leveled by positioning both eyes at the same height as slit lamp light. All patients followed the same pre- and postoperative medications with topical levofloxacin drops and dexamethasone 0.1%. All patients with relevant posterior capsule opacification underwent Nd:YAG (neodymium-doped yttrium aluminum garnet) laser capsulotomy. 

### 2.4. Examination

Three to nine months following surgery, all patients were checked using ETDRS charts for uncorrected distance visual acuity (UDCVA) and best-corrected distance visual acuity (BCDVA) using ETDRS charts. Testing was performed at 5 m with liquid display charts calibrated to 5 m ETDRS letters. In addition, uncorrected near visual acuity (UNVA) at 40 cm and best-corrected near visual acuity (BCNVA) at 40 cm were assessed using Snellen charts in logMAR units. Tear break-up time (TBUT) was measured 3 times using the observational ophthalmoscopy method, and a mean measurement was calculated. TBUT was performed by installation of ophthalmic quality fluorescein dye of the BioGlo, one drop into tear film manufactured by Hub Pharmaceuticals, Plymouth, USA. Poor tear film makes patients with a range of vision intraocular lenses implanted more susceptible to visual disturbances [20]. Refractive error and intraocular pressure measurements were part of all ophthalmological examinations. The evaluation included ocular biomicroscopy with peripheral and central funduscopy. Aberrations and refraction of all patients were measured using the aberrometer iTrace (Tracey Technologies Corporation, Houston, TX, USA, software version 6.2.0) with physiological pupil size. The iTrace uses a ray tracing principle with sequential scanning of grid-spaced 256 near-infrared beams through the eye.

The iTrace measurements were performed after dark room mydriasis (scotopic pupil size) and a fixation target projected to infinity. The measurement takes less than 200 ms. The total and internal aberrations were calculated by subtracting measurements by the Placido-based topographer mounted on the iTrace corneal wavefront aberrations from those of the entire eye measured by the ray tracing aberrometer [21]. Typical images of aberrometric retinal spot diagrams of treated eyes with studied lenses are shown in Figure 2 and Figure 3. TraceRef (SEQ)—spherical equivalent refraction was measured by ray tracing. Low-order aberrations, i.e., Defocus and Astigmatism are additionally shown in microns as iTrace machine delivers the measurement. Coma measured by iTrace aberrometry included both third-order vertical and horizontal coma (Z_3_^−1^ and Z_3_^1^), as well as Trefoil included both third-order vertical and oblique trefoil (Z_3_^−3^ and Z_3_^3^). Spherical measurements included only the Primary Spherical Aberration (Z_4_^0^). 

### 2.5. Spherical Equivalent Refractive Accuracy

Statistical analyses were performed on all the data collected in an Excel database (version 2019, Microsoft, Redmond, WA, USA) using the IBM SPSS Statistics 27 package and Statistica v.13.3 program (TIBCO, Palo Alto, CA, USA). The basic descriptive statistics were analyzed using the Shapiro–Wilk tests and the Spearman rho correlation analysis. To analyze data collected from both eyes, a nested ANOVA test for bilaterality was performed. *p* values less than 0.05 were considered statistically significant, followed by Post hoc Dunn tests with Bonferroni correction for the measurements.

First, the presence of outliers was checked by transforming the measured results into standardized values. The basic descriptive statistics of the studied variables were also calculated. 

All distributions were tested for normality using the Shapiro–Wilk test. In the case of normality, the parametric *t*-test for unpaired samples or Pearson was used, and in the case of non-normality, the nonparametric Mann–Whitney-U test or Spearman was considered.

The Kruskal–Wallis test was performed for difference presence following surgery. 

## 3. Results

Forty-two eyes of twenty-two patients assessed by iTrace were included in group 1 with an age range from 33 to 84 years old (56.5 ± 11.5 years, mean ± standard deviation) with the Lentis Mplus15 lens. Ninety-one eyes of fifty-three patients aged 31 to 79 years (59.2 ± 13.1 years, mean ± standard deviations) were included in group 2, with the Lentis Mplus30 lens. The control group consisted of 107 eyes of 56 patients aged 27 to 73 years (46.9 ± 10.9 years, mean ± standard deviation). Gender distribution (0.31) was not significantly different between groups while age was significant between controls and Lentis Mplus 15 *p* < 0.001 and M30 *p* < 0.001 while it was not significant between both IOL groups (*p* = 0.22).

The results of the basic descriptive statistics analysis are presented by group in Table 1.

### 3.1. Visual Acuity and Refraction

Overall, most eyes in Group 1 and 2 had a final UDVA and CDVA that was higher than 80 EDTRS letters (74% and 85% respectively) (Figure 4), while in these two groups no eye lost lines of CDVA, 22% gained one or more lines of CDVA (10% gained one line, 7% gained two lines, 5% gained three or more lines on ETDRS chart) (Figure 5). Finally, the postoperative refractive spherical equivalent of ±1.00 Dpt was achieved in 72% of eyes, and 45% of all eyes achieved a postoperative refractive spherical equivalent of ±0.50 Dpt (Figure 6).

In the next part of the analysis, we checked whether the values of HOA Total, Defocus, Coma, Spherical, Trefoil, Astigmatism, and Secondary Astigmatism measurements differed between preoperative and postoperative subjects who had the Lentis Mplus15 or Mplus30 lenses (Figure 7).

The analysis showed both study groups yielded statistically significant differences in all parameters except for Defocus. Post hoc Dunn tests with Bonferroni correction were performed for the measurements. The analysis allowed us to state that in terms of the HOA Total, Coma, Trefoil, and Secondary Astigmatism measurements, there were statistically significant differences between patients with the Lentis Mplus30 lens and the other two groups. In the case of HOA Total, Coma, Trefoil, and Secondary Astigmatism, higher measurement values were obtained in the postoperative group with the Mplus30 lens compared to the second postoperative group (Mplus15) and the control group. The difference in the value of Secondary Astigmatism between the control and postoperative groups with the Mplus30 lens was significant at *p* = 0.002, and the remaining ones at *p* < 0.001. Concerning spherical equivalent refraction error, the group with the Lentis Mplus15 lens obtained significantly higher values than the control group (difference at the level of *p* = 0.016) and the postoperative group with the Lentis Mplus30 lens (difference at the level of *p* < 0.001).

On the other hand, the control group did not differ significantly from the postoperative group in which the Mplus30 lens was used. Astigmatism in the control group was significantly higher (*p* = 0.028) than in the postoperative group, in which the Mplus15 lens was used. The differences between the remaining pairs (Mplus30 vs. control group) of groups were not statistically significant.

In the next part of the analysis, it was checked whether the UDVA, UCNVA, and TBUT measurements differed between patients before and after surgery with both study groups. For this purpose, the Kruskal–Wallis test was performed again (Table 2).

The analysis showed the differences in all variables under test, comparing both study groups and the controls. Dunn’s post hoc tests with Bonferroni correction were performed to check in which groups there were differences. The results indicated that the control group differed statistically in the UDVA, UNVA, and TBUT measurements from both postoperative groups (*p* < 0.001 each time). Preoperative UDVA was lower before surgery while UNVA in logMAR was higher after the procedure as well as TBUT. On the other hand, for the postoperative exams, there was no statistically significant difference in UDVA, UNVA, and TBUT measurements.

Then, we checked whether, by postoperative exam of patients with the Mplus15 lens and patients with the Mplus30 lens, the levels of the variables HOA Total and LOA Total were associated with the variable TraceRef (SEQ). For this purpose, the Spearman rho correlation analysis was performed (Table 3).

Only the LO aberrations in the control group and in the Mplus30 reached statistical significance in the Spearman correlation.

### 3.2. Relationship between Visual Acuity and the Tear Film Break-Up Time

For the next step, we checked whether the TBUT shows some impact on the BCDVA or BCNVA in any study or control group. For this purpose, Spearman’s rho correlation analysis was performed (Table 4).

Only in the Mplus15 did the Spearman correlation reach statistical significance with the UNVA. All other relationships were statistically insignificant.

## 4. Discussion

The present study compares optical aberrations between two types of intraocular lenses that differ in near-addition power. The efficacy of the MF IOLs for visual distance achievements is justified. This finding is consistent with other studies of multifocal lens designs [22,23,24]. Patients with the Mplus15 lens achieved better UDVA, and patients with Mplus30 had better UNVA, but no statistical difference was noticed. The authors studied the distance and near function of patients with both lenses. Because intraocular lenses with two different additions were used (+1.5 versus +3.0 dpt), an important limitation of the study is not considering uncorrected intermediate visual acuity assessment. 

The restoration of the accommodation function by pseudophakic pseudoaccommodation following a refractive procedure or cataract surgery mainly depends on the type of multifocal intraocular lens implanted. Other factors influencing good vision function are surgical success elements such as lens centration intraoperative and effective postoperative, tilt, capsulorhexis regularity, corneal incision impassivity, wound healing effects, residual refraction errors, posterior capsule opacification (PCO), post-surgical tear film pathologies, and wavefront abnormalities.

The studied MF IOLs were designed to achieve spectacle-free functional distance and near vision [1,25]. However, the above factors can contribute to a substantial increase in HOA causing inadequate uncorrected distance visual acuity.

The rotationally asymmetric MF IOLs have been extensively evaluated for their efficacy in vision restoration, but visual disturbance in terms of higher-order aberrations may limit these lenses’ potential benefit [26]. The other multifocal intraocular lens technology, the diffractive type, may cause photic phenomena such as halos, glares, and decreased contrast sensitivity [27]. However photic phenomena are not restricted to diffractive technology. The Mplus30 and Mplus15 lenses are the first commercially available rotationally asymmetrical multifocal lenses. Multiple studies have shown excellent visual outcomes with reasonable distance and near function [10,28,29,30].

The limitation of UNVA for the Mplus15 lens group is associated with the +1.5 dpt addition, which is a lower optical power than Mplus30. However, adaptation to spectacle independence with these lenses may play an essential role in good overall function in most light conditions and distances. Neuroadaptation as a process requires time and the activity of cortical areas dedicated to attention (frontoparietal circuits), learning and cognitive control (cingulate), and task goals (caudate) [31]. Improvement in UNVA is statistically significant for both groups compared to the preoperative control group. 

Spherical equivalent error achieved in both groups could be affected by the intraocular lens calculation formula precision and accuracy. Those depend on the reliability of preoperative measurements, achieving the desired effective lens position and possible postoperative fluctuations in corneal power [32,33,34,35,36].

Mplus30 shows a significantly higher HOA compared to the +1.5-diopter addition lens Mplus15. As expected, these findings are consistent with previously published studies in terms of limiting the optical quality with the use of a larger addition for the rotationally asymmetric IOL [37,38].

HOA aberrations depend on pupil size. We used physiological pupil size under scotopic light conditions; however, the patients’ pupil reactions and functions differ, depending on eye comorbidities, medication exposure, age, and heritability of genetic determination. The limitation of our study was the difference in pupil size between the groups. By this difference, the influence on HOA aberrations could be noticed and could be a subject for further analysis. Most probably the pupil size should be of interest to ophthalmic surgeons when choosing a multifocal lens. 

The TBUT test, a clinical tear film assessment, is usually disturbed following lens surgery and is associated with discomfort, blurry vision, and photopic phenomena. Dry eye may be induced or increased by reducing the corneal sensitivity due to cataract incisions [39,40]. There were no statistical differences noted in TBUT between the groups. However, there was an observed correlation for extended TBUT in Mplus15 patients with insufficient UNVA. The worse the UNVA, the higher the TBUT.

In our study, one of the limitations is that the aberration data analyzed considered the total data, not separately for the internal and anterior surface corneal data. Once achieving separate anterior surface corneal data, most desirably with no statistical difference, it would be possible to obtain precise data of the studied multifocal influence on the eye’s aberrations without the anterior surface corneal bias.

Another limitation is that human polychromatic vision is assessed by an aberrometer performing monochromatic measurements.

An important limitation of the study is the heterogeneity of the groups concerning the age of patients and the standardization of the aberrations and ocular parameters within the studied groups.

The authors of the study used a nested ANOVA test to overcome the problem of the possibility of obtaining smaller *p*-values in the same group from the bilateral eyes examined. However, there are other specific statistical methods such as mixed linear models, the Bootstrap or generalized estimating equations, which could be used [41].

Patients with the low-addition lens—the Mplus15—had lower total HOA and LOA and higher spherical aberrations than patients with the Mplus30 lens. In the future, the analysis of contrast sensitivity could allow for a more precise overall assessment of the quality of vision in patients with MF IOL implanted. In addition, a detailed visual performance and quality of vision questionnaire survey may contribute to a better prognosis for multifocal intraocular lens use in patients. The analysis of both eyes can be seen as a limitation; however, it is possible to use a hierarchical statistical test such as a nested ANOVA to overcome two-eye bias.

The decision on the lens choice should be based on, but not limited to, factors studied. The clinical recommendation should be in preference for the use of the Mplus15 lens over the Mplus30 lens.

## 5. Conclusions

In conclusion, the Mplus15 and Mplus30 lenses both provide comparable, satisfactory near and distance visual results; however, the implantation of the Mplus30 lens results in higher total HOA and LOA compared to the Mplus15 lens. Thus, counseling patients before surgery about potential photic phenomena is necessary. 

## Figures and Tables

**Figure 1 jcm-13-00239-f001:**
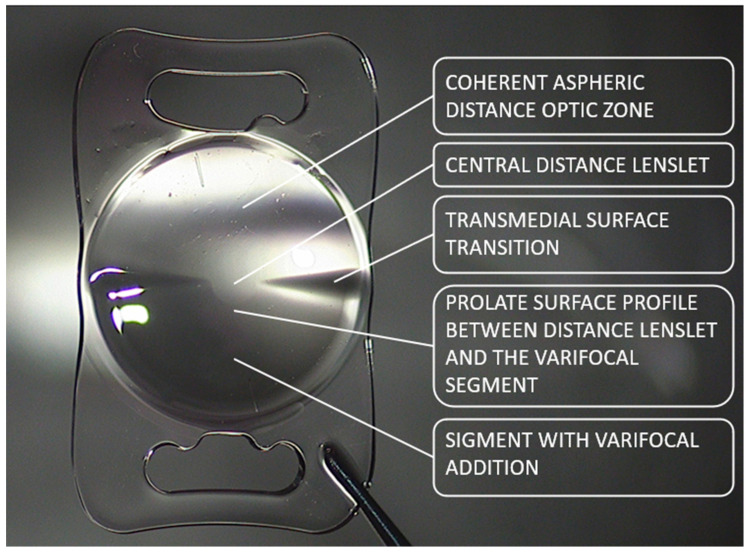
The Mplus15 and Mplus30 lens schematic with the asymmetric lens design.

**Figure 2 jcm-13-00239-f002:**
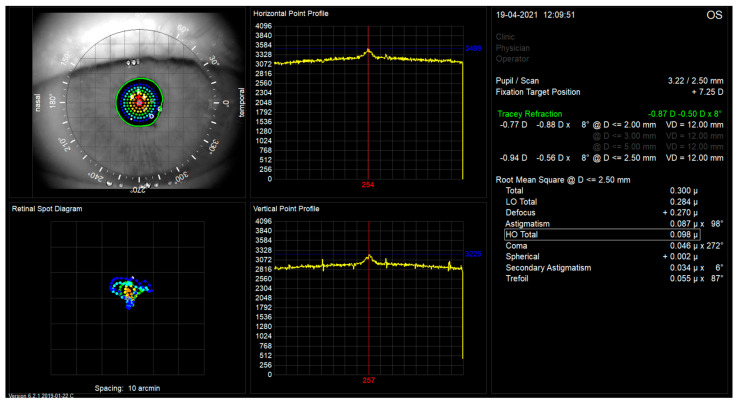
Image of typical retinal spot diagram for patient with Mplus15 lens implanted.

**Figure 3 jcm-13-00239-f003:**
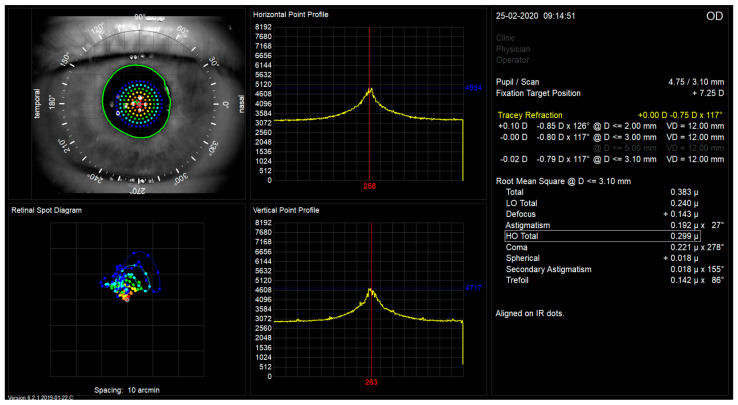
Image of typical retinal spot diagram for patient with Mplus30 lens implanted.

**Figure 4 jcm-13-00239-f004:**
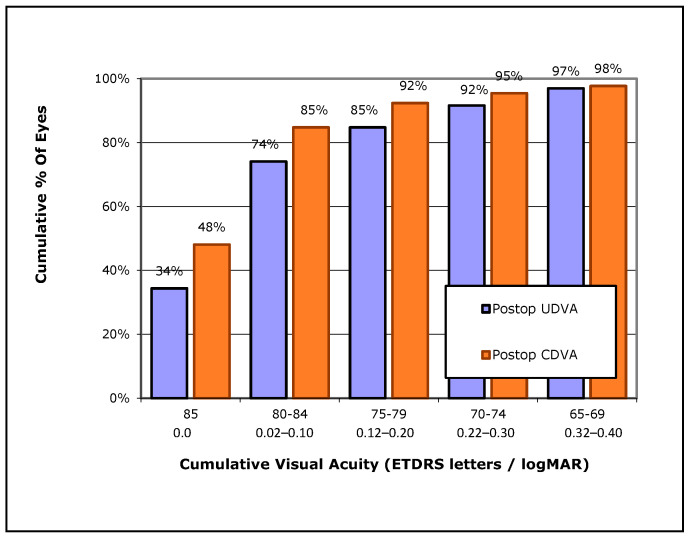
Cumulative percentage of eyes with a given uncorrected distance visual acuity (UDVA) vs. corrected distance visual acuity (CDVA) in all the cases included in Groups 1 and 2.

**Figure 5 jcm-13-00239-f005:**
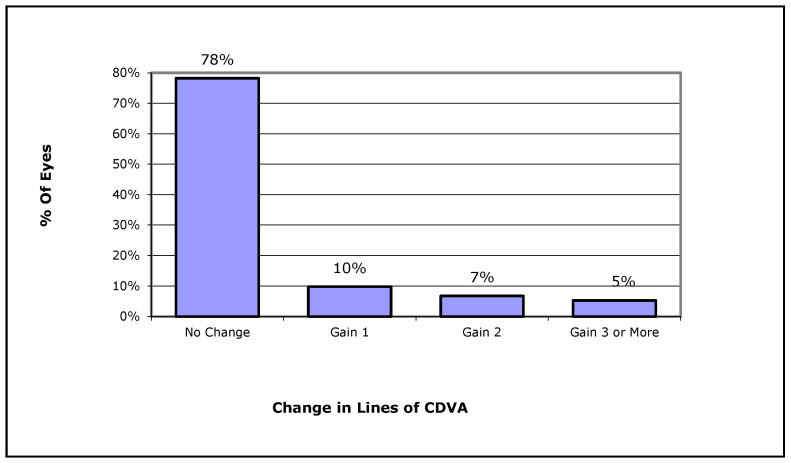
Change in corrected distance visual acuity (CDVA) in all cases included in Groups 1 and 2.

**Figure 6 jcm-13-00239-f006:**
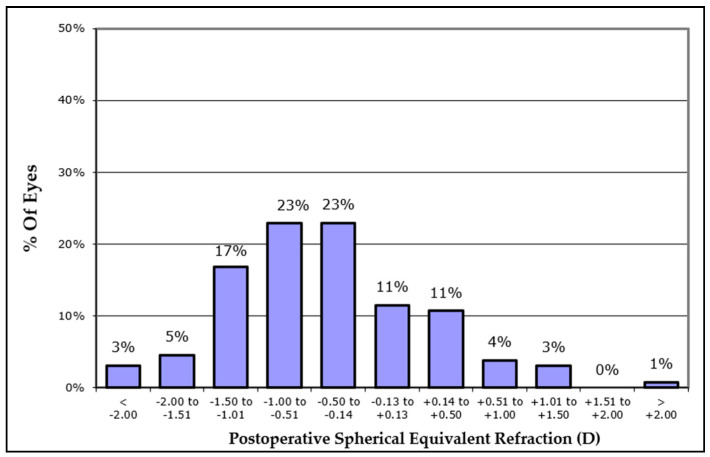
Cumulative postoperative spherical equivalent refraction in all cases included in Groups 1 and 2.

**Figure 7 jcm-13-00239-f007:**
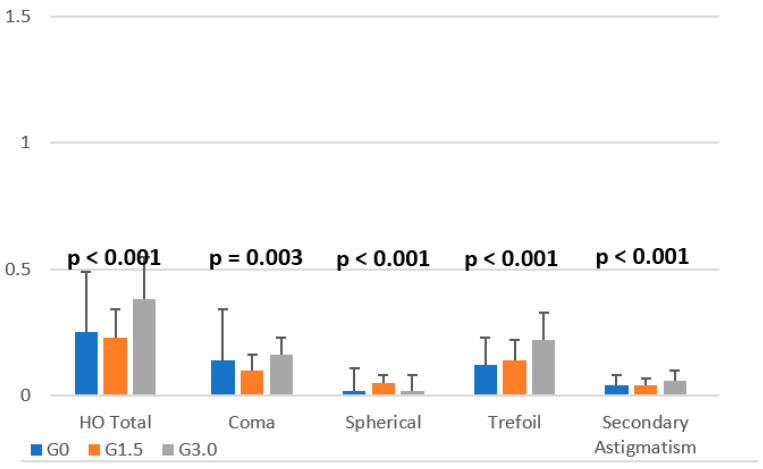
Differentiation of HOA Total, Coma, Spherical, Trefoil, and Secondary Astigmatism parameters depending on the group. Annotation. HOA—high order; G0—patients before surgery (control group); G1.5—postoperative patients w/Lentis Mplus15 lens; G3.0—postoperative patients w/Lentis Mplus30 lens.

**Table 1 jcm-13-00239-t001:** Basic descriptive statistics of the recorded parameters, taking into account the division into groups.

Tested Parameter	Control Group (Preoperative Measurement)		Patients w/Lentis Mplus15 Lens		Patients w/Lentis Mplus30 Lens	
	Means	SD	Mean	SD	Mean	SD
HOA Total in µm	0.25	0.24	0.23	0.11	0.38	0.17
LOA Total in µm	1.91	1.62	0.39	0.33	0.45	0.31
Defocus in µm	0.5	2.28	0.07	0.4	0.22	0.39
Coma in µm	0.14	0.2	0.1	0.06	0.16	0.07
Spherical in µm	0.02	0.09	0.05	0.03	0.02	0.06
Trefoil in µm	0.12	0.11	0.14	0.08	0.22	0.11
Astigmatism in µm	0.58	0.71	0.25	0.18	0.28	0.14
Secondary Astigmatism in µm	0.04	0.04	0.04	0.03	0.06	0.04
UDVA EDTRS Letters	55.57	23.96	81.24	5.55	79.81	8.24
BCDVA EDTRS Letters	81.44	7.32	81.74	4.62	82.03	7.12
UNVA logMAR	0.35	0.29	0.05	0.07	0.03	0.1
TBUT in s	8.31	2.42	5.62	1.48	5.1	1.96
TraceRef SEQ in dpt	−1.28	4.65	−0.25	0.65	−0.6	0.87
DEQ in dpt			0.81	0.40	1.21	0.68
Pupil diameter in mm	3.85	1.04	3.19	0.84	2.89	0.70

Annotation: UDVA—uncorrected distance visual acuity; BCDVA—best-corrected distance visual acuity; UNVA—uncorrected near visual acuity; TBUT—tear break-up time; HOA—higher-order aberrations; LOA—lower-order aberrations; SEQ—spherical equivalent; DEQ—defocus equivalent.

**Table 2 jcm-13-00239-t002:** Differentiation of UDVA, UCNVA, and TBUT parameters depending on the group.

	Group	*n*	*M*	*SD*	*Me*	Average Rank	*H*	*df*	*p*
UDVA EDTRS letters	G0	103	55.57	23.96	58.90	73.39	80.95	2	<0.001
	G1.5	42	81.24	5.55	83.00	155.86			
	G3.0	91	79.81	8.24	84.00	152.31			
UNVA logMAR	G0	64	0.35	0.29	0.20	142.05	71.74	2	<0.001
	G1.5	42	0.05	0.07	0.00	87.52			
	G3.0	91	0.03	0.10	0.00	74.02			
TBUT in s	G0	107	8.31	2.42	8.00	165.07	81.15	2	<0.001
	G1.5	42	5.62	1.48	6.00	94.56			
	G3.0	91	5.10	1.96	5.00	80.06			

Annotation. G0—patients before surgery (control group); G1.5—postoperative patients w/Mplus15 lens; G3.0—postoperative patients w/Mplus30 lens; UDVA—uncorrected distance visual acuity; UNVA—uncorrected near visual acuity; TBUT—tear break-up time. *M*—mean, *SD*—standard deviation, *Me*—Median.

**Table 3 jcm-13-00239-t003:** Correlation between HOA Total and LOA Total and SEQ in individual study groups.

Patients before surgery (control group)	HOA Total (*n* = 107)	*rho* Spearman significance	0.13
			0.187
	LOA Total (*n* = 107)	*rho* Spearman significance	−0.24
			0.014
Patients with Mplus15	HOA Total (*n* = 42)	*rho* Spearman significance	0.14
			0.373
	LOA Total (*n* = 42)	*rho* Spearman significance	−0.15
			0.336
Patients with Mplus30	HOA Total (*n* = 91)	*rho* Spearman significance	0.03
			0.762
	LOA Total (*n* = 91)	*rho* Spearman significance	−0.46
			<0.001

**Table 4 jcm-13-00239-t004:** Correlation between BCDVA and BCNVA and TBUT in particular study groups.

			TBUT (s)
Patients before surgery (control group)	BCDVA (*n* = 103)	*rho* Spearman significance	−0.01
			0.967
	BCNVA (*n* = 64)	*rho* Spearman significance	−0.10
			0.300
Patients with Mplus15	BCDVA (*n* = 42)	*rho* Spearman significance	0.05
			0.77
	BCNVA (*n* = 42)	*rho* Spearman significance	0.18
			0.26
Patients with Mplus30	BCDVA (*n* = 91)	*rho* Spearman significance	0.13
			0.22
	BCNVA (*n* = 91)	*rho* Spearman significance	−0.16
			0.118

## Data Availability

The data presented in this study are available on request from the corresponding author.

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
