# Peer review of "Comparative Analysis of the Visual, Refractive and Aberrometric Outcome with the Use of 2 Intraocular Refractive Segment Multifocal Lenses"

_jcm, 2023, doi:10.3390/jcm13010239_

Round 1
Reviewer 1 Report
Comments and Suggestions for Authors
The manuscript titled "Comparative Analysis of Visual Outcomes and Optical Aberrations with Mplus15 and Mplus30 Multifocal Intraocular Lenses" presents a comprehensive investigation into the visual performance and aberrations associated with different multifocal intraocular lenses (IOLs). The authors conducted a well-structured study, employing Spearman correlation analyses and evaluating parameters such as total higher-order aberrations (HOA), tear film break-up time (TBUT), and visual acuity.
The results indicate that patients with the Mplus15 lens exhibited better uncorrected distance visual acuity (UDVA), while those with the Mplus30 lens showed improved uncorrected near visual acuity (UNVA). The study adeptly addresses various factors influencing visual function post-refractive surgery, including lens centration, corneal incision, and post-surgical tear film pathologies. Moreover, the authors discuss the impact of rotationally asymmetric multifocal lenses on higher-order aberrations, emphasizing the potential for visual disturbances.
The limitations of the study, such as the heterogeneity of patient groups and the assessment of polychromatic vision through monochromatic measurements, are duly acknowledged. Additionally, the correlation between tear film break-up time and UNVA, along with the observed differences in total HOA between Mplus15 and Mplus30 lenses, adds valuable insights to the multifocal lens literature.
The comprehensive review of relevant literature and proper citation of previous studies strengthen the manuscript's scientific foundation. However, addressing the limitations by considering separate anterior surface corneal data and employing a hierarchical statistical test for two-eye bias would further enhance the robustness of the study.
In conclusion, this paper significantly contributes to the understanding of visual outcomes associated with different multifocal intraocular lenses, shedding light on potential factors influencing aberrations and visual acuity. The thorough methodology, clear presentation of results, and identification of study limitations demonstrate the authors' commitment to scientific rigor. Overall, this work holds importance for clinicians considering multifocal IOL options and provides a basis for future investigations in the field.
Author Response
RESPONSE TO REVIEWER 1 COMMENTS
Thank you very much for taking the time to review this manuscript. Please find the detailed responses below and the corresponding revisions/corrections highlighted/in track changes in the re-submitted files.
Comments and Suggestions for Authors
The manuscript titled "Comparative Analysis of Visual Outcomes and Optical Aberrations with Mplus15 and Mplus30 Multifocal Intraocular Lenses" presents a comprehensive investigation into the visual performance and aberrations associated with different multifocal intraocular lenses (IOLs). The authors conducted a well-structured study, employing Spearman correlation analyses and evaluating parameters such as total higher-order aberrations (HOA), tear film break-up time (TBUT), and visual acuity.
The results indicate that patients with the Mplus15 lens exhibited better uncorrected distance visual acuity (UDVA), while those with the Mplus30 lens showed improved uncorrected near visual acuity (UNVA). The study adeptly addresses various factors influencing visual function post-refractive surgery, including lens centration, corneal incision, and post-surgical tear film pathologies. Moreover, the authors discuss the impact of rotationally asymmetric multifocal lenses on higher-order aberrations, emphasizing the potential for visual disturbances.
The limitations of the study, such as the heterogeneity of patient groups and the assessment of polychromatic vision through monochromatic measurements, are duly acknowledged. Additionally, the correlation between tear film break-up time and UNVA, along with the observed differences in total HOA between Mplus15 and Mplus30 lenses, adds valuable insights to the multifocal lens literature.
The comprehensive review of relevant literature and proper citation of previous studies strengthen the manuscript's scientific foundation. However, addressing the limitations by considering separate anterior surface corneal data and employing a hierarchical statistical test for two-eye bias would further enhance the robustness of the study.
In conclusion, this paper significantly contributes to the understanding of visual outcomes associated with different multifocal intraocular lenses, shedding light on potential factors influencing aberrations and visual acuity. The thorough methodology, clear presentation of results, and identification of study limitations demonstrate the authors' commitment to scientific rigor. Overall, this work holds importance for clinicians considering multifocal IOL options and provides a basis for future investigations in the field.
Response: Thank you very much for taking the time to review this manuscript. We do agree that not considering separate anterior surface corneal data and employing a hierarchical statistical test for two eyes bias is an important limitation. We have modified the discussion section accordingly. Thank you.
Reviewer 2 Report
Comments and Suggestions for Authors
Comparative Analysis of the Visual, Refractive and Aberrometric Outcome with the Use of 2 Intraocular Refractive Segment Multifocal Lenses.
In page 1, in “Abstract”, line 5, it reads. “…were measured at scotopic pupil size in 42 eyes of patients implanted..”.
According to the description in “Examination” section, pharmacological mydriasis was used, therefore this is not a “scotopic pupil size”, which would be a naturally dilated pupil in low light conditions. This is a pharmacologically dilated pupil.
Therefore it should read: “…were measured through a pharmacologically dilated pupil in 42 eyes of patients implanted..”.
According to the descr
In page 1, in “Abstract”, line 7, it reads. “…and 107 eyes of control group. Results: No significant differences were noticed between the examined groups concerning UCDVA, UCNVA, and tear break-up time (p < 0.001).”
Comment:
Age-matched?
In page 1, in “Introduction”, first paragraph, first line, it reads: “Cataract surgery or refractive lens exchange with the implantation of multifocal intraocular lens (MF IOL) allows for good visual performance and spectacle independence [1]. The aim of the surgery is to provide rapid and complete visual rehabilitation with marginal postoperative refractive errors [2].”
Comment: Since the term “multifocal intraocular lenses” includes both bifocal and trifocal platforms, the statement is too broad and succinct. As several newer trifocal models have been marketed and successfully implanted in many patients, they should be at least mentioned and articles about their performance cited.
Therefore, consider modifying to: “Cataract surgery or refractive lens exchange with the implantation of multifocal intraocular lens (MF IOL) allows for good visual performance and spectacle independence, particularly for the newer trifocal platforms (AtLisa tri 839 MP, Panoptix, Fine Vision, among others), which let the patients to have good unaided near, intermediate and far vision [1] (Breyer 2020; Niazi 2023; Galvis 2022; LubiÅ„ski2020; Ang 2023; Karam 2023). The aim of the surgery is to provide rapid and complete visual rehabilitation with marginal postoperative refractive errors, and good enough uncorrected visual acuity at the three distances. In cases of significant residual ametropia, additional photorefractive procedures may be needed [2].
New references to be cited:
Breyer DRH, Beckers L, Ax T, Kaymak H, Klabe K, Kretz FTA. Aktuelle Übersicht: multifokale Linsen und Extended-Depth-of-Focus-Intraokularlinsen [Current Review: Multifocal Intraocular Lenses and Extended Depth of Focus Intraocular Lenses]. Klin Monbl Augenheilkd. 2020;237(8):943-957. doi:10.1055/a-1111-9380
Niazi S, Gatzioufas Z, Dhubhghaill SN, et al. Association of Patient Satisfaction with Cataract Grading in Five Types of Multifocal IOLs [published online ahead of print, 2023 Oct 27]. Adv Ther. 2023;10.1007/s12325-023-02698-5. doi:10.1007/s12325-023-02698-5.
Galvis V, Escaf LC, Escaf LJ, et al. Visual and satisfaction results with implantation of the trifocal Panoptix® intraocular lens in cataract surgery. J Optom. 2022;15(3):219-227. doi:10.1016/j.optom.2021.05.002
Lubiński W, Podborączyńska-Jodko K, Kirkiewicz M, Mularczyk M, Post M. Comparison of visual outcomes after implantation of AtLisa tri 839 MP and Symfony intraocular lenses. Int Ophthalmol. 2020;40(10):2553-2562. doi:10.1007/s10792-020-01435-z
Ang RET. Long-term trifocal toric intraocular lens outcomes in Asian eyes after cataract surgery. J Cataract Refract Surg. 2023;49(8):832-839. doi:10.1097/j.jcrs.0000000000001195
Karam M, Alkhowaiter N, Alkhabbaz A, et al. Extended Depth of Focus Versus Trifocal for Intraocular Lens Implantation: An Updated Systematic Review and Meta-Analysis. Am J Ophthalmol. 2023;251:52-70. doi:10.1016/j.ajo.2023.01.024.
In page 1, in “Introduction”, first paragraph, line 4, it reads: “Studies have reported excellent visual outcomes and low occurrence of photic phenomena following the use of multifocal rotational asymmetric intraocular lenses [3].”
Comment:
It is important that the reader has the concept of this different approach, rotational asymmetric intraocular lenses, and that unlike trifocal models, it does not yield good uncorrected intermediate vision.
Therefore, consider modifying to: “An additional approach has been the use of rotational asymmetric intraocular lenses. These platforms exhibit refractive rotational asymmetry unlike the standard bifocal or trifocal intraocular lenses, which are rotationally symmetric and based on the principles of diffraction and refraction. Studies have reported excellent visual outcomes and low occurrence of photic phenomena following the use of these multifocal rotational asymmetric intraocular lenses. The amount of addition is variable (usually +3.00 D for near vision, or +1.50 D for intermediate vision models). [3] (McNeely 2016; Oshika, 2029).”
Additional reference:
McNeely RN, Pazo E, Spence A, Richoz O, Nesbit MA, Moore TC, Moore JE. Comparison of the visual performance and quality of vision with combined symmetrical inferonasal near addition versus inferonasal and superotemporal placement of rotationally asymmetric refractive multifocal intraocular lenses. J Cataract Refract Surg. 2016 Dec;42(12):1721-1729. doi: 10.1016/j.jcrs.2016.10.016. PMID: 28007103.
Oshika T, Arai H, Fujita Y, Inamura M, Inoue Y, Noda T, Miyata K. One-year clinical evaluation of rotationally asymmetric multifocal intraocular lens with +1.5 diopters near addition. Sci Rep. 2019 Sep 11;9(1):13117. doi: 10.1038/s41598-019-49524-z. PMID: 31511557; PMCID: PMC6739307.
In page 1, in “Introduction”, first paragraph, line 6, it reads: “Aberrations refer to the differences in the optical path-lengths (actual minus ideal wavefront achieved in the optical system). Optical aberrations of a specific visible light wavelength travelling through an optical system are generally classified into several categories: spherical refractive error (defocus), cylindrical refractive error, spherical aberration, coma, and other higher-order aberrations (HOA) [4]. Light scattering and HOA due to the refractive surface of the intraocular lens may lead to poor retinal image quality [5].”
Comment:
The statements are confusing and incomplete. Consider modifying to:
“Aberrations refer to differences in optical path lengths, which represent the deviation between the actual and ideal wavefront achieved in an optical system. Optical aberrations for a specific visible light wavelength traveling through an optical system are generally classified into two broad categories. The first category includes lower order aberrations, which account for most of the degradation in image quality. These lower order aberrations comprise spherical refractive error (defocus) and cylindrical refractive error. The second category is higher order aberrations (HOA), encompassing spherical aberration, coma, and others. When significantly high, HOA can impact the quality of vision and, notably, cannot be corrected with spectacles or contact lenses. [4]. Light scattering can also, along with HOAs, further degrade retinal image quality [5].”
In page 1, in “Introduction”, first paragraph, line 12, it reads: “However, the optical aberrations may differ depending on the amount of near addition. The study aimed to compare visual, refractive, and aberrometric outcomes following surgery with the use of two refractive central-sparing rotational asymmetric multifocal lenses with near vision addition of +1.5 and +3.0 dpt.”
Comment.
The wording should be revised. In addition, it is important to mention that clinical objective measurements of IOLs performance (v.gr. aberrometry and double-pass optical visual quality analysis) have been questioned as to their real ability to evaluate the clinical performance of these platforms (Charman 2008; Fernández 2022; Vega 2023), and that they may be just a rough approximation of the in vivo performance of the IOLs as perceived by the patients. However, they can yield some useful information in some cases (D'Oria 2023). This issue should be mentioned both in “Introduction” and in more detail in “Discussion” section.
Therefore, consider modifying to:
“Optical aberrations may differ depending on many features of the introcular lenses, including the amount of near addition. Although, currently there is not a consensus about the real ability of aberrometers to evaluate the clinical performance of multifocal introcular lenses (Charman 2008; Fernández 2022; Vega 2023), it seems that they can provide useful information (D'Oria 2023). The present study aimed to compare visual, refractive, and aberrometric outcomes following surgery with the use of two refractive central-sparing rotational asymmetric multifocal intraocular lenses with two different amounts of near vision addition, specifically +1.5 versus +3.0 dpt.”
Additional References:
Charman WN, Montés-Micó R, Radhakrishnan H. Problems in the measurement of wavefront aberration for eyes implanted with diffractive bifocal and multifocal intraocular lenses. J Refract Surg. 2008 Mar;24(3):280-6. doi: 10.3928/1081597X-20080301-10. PMID: 18416263.
Fernández J, Rocha-de-Lossada C, Rodríguez-Vallejo M. Objective Optical Quality With Multifocal Intraocular Lenses Should Stop to Be Used or Cautiously Interpreted. Asia Pac J Ophthalmol (Phila). 2022;11(6):569. Published 2022 Nov 1. doi:10.1097/APO.0000000000000502.
Vega F, Faria-Ribeiro M, Armengol J, Millán MS. Pitfalls of Using NIR-Based Clinical Instruments to Test Eyes Implanted with Diffractive Intraocular Lenses. Diagnostics (Basel). 2023 Mar 27;13(7):1259. doi: 10.3390/diagnostics13071259. PMID: 37046477; PMCID: PMC10093131.
D'Oria F, Scotti G, Sborgia A, Boscia F, Alessio G. How Reliable Is Pyramidal Wavefront-Based Sensor Aberrometry in Measuring the In Vivo Optical Behaviour of Multifocal IOLs? Sensors (Basel). 2023 Mar 28;23(7):3534. doi: 10.3390/s23073534. PMID: 37050594; PMCID: PMC10099035.
Page 2, first paragraph following “Patients” subtitle, line 4. It reads: “The control group consisted of patients who were qualified for refractive surgery.”
Comment: It is very important to clarify here what type of refractive surgery (because indeed refractive lens exchange is a type of refractive surgery). I suppose the authors meant excimer laser corneal refractive surgery.
Page 4, first paragraph, line 3. It reads: “In addition, uncorrected near visual acuity (UNVA) at 40 cm and best-corrected near visual acuity (BCNVA) at 40 cm were assessed using logMAR charts”
Comment:
Considering that models with two different additions were used (+1.5 versus +3.0 dpt), both near and intermediate uncorrected visual acuity should have been evaluated. If not, this weakness must be acknowledged.
Page 4, second paragraph, line 1. It reads: “The iTrace measurements were performed after pharmacological mydriasis (scotopic pupil size) and a fixation target projected to infinity.”
Comment: A pharmacologically dilated pupil is not a “scotopic pupil”, which would be a naturally dilated pupil in darkness, without medication.
It is crucial to specify the mydriatic substance applied (such as phenylephrine, tropicamide, or cyclopentolate) and the corresponding doses used. Another critical consideration is providing information on the diameter of the pupil used for aberrometric measurements. Aberrometry is linearly related to the measurement diameter, making this detail fundamental. If different diameters were employed for different eyes, it is essential to thoroughly review the data, establish a consistent diameter (commonly 6.00 mm), and then redo the analysis with the obtained values. Therefore in Table 1 it should be specified the diameter used to determine the aberrometric parameters.
In Table 1, the head of the control group column, should be modified to “Control Group (preoperative measurement)”.
In Table 1, it is necessary to clarify whether “Coma” includes both, third order vertical and horizontal coma (Z3-1 and Z31) , whether “Trefoil” includes both, third order vertical and oblique trefoil (Z3-3 and Z33), and whether Spherical aberration includes only the Primary Spherical Aberration (Z40) or also the secondary spherical aberration (Z60).
Usually lower order aberrations, i.e. Defocus and Astigmatism, are not shown in microns, since they can be simply clinically determined in diopters, and used to calculate both Spherical Equivalent and Defocus equivalent. Therefore, I consider the information in microns is superfluous, and may be confusing for the reader.
The Uncorrected Distance Visual Acuity (UDVA) in the control group, being entirely contingent upon refractive error, provides limited meaningful information and is recommended for deletion. Instead, a more relevant comparison could be established by including the preoperative Distance Corrected Visual Acuity of the control group. This would allow for an effective comparison with the postoperative UDVA of eyes implanted with the Lentis Mplus intraocular lenses.
Page 5. Second paragraph following “Spherical equivalent refractive accuracy”, line 3. It reads:” The results of the analysis are presented by group in Table 1.”
Comment: This information should not be included in this section (“Methods”), but in the “Results” section.
In addition, it would be interesting and useful for the reader to calculate and analyze the “Defocus Equivalent” of the eyes included in the study.
Defocus Equivalent = Absolute Value Spherical Equivalent + (Absolute Value Cylinder/2)
Holladay JT, Moran JR, Kezirian GM. Analysis of aggregate surgically induced refractive change, prediction error, and intraocular astigmatism. J Cataract Refract Surg. 2001 Jan;27(1):61-79. doi: 10.1016/s0886-3350(00)00796-3. PMID: 11165858.
Page 6, first paragraph following “Visual Acuity and Refraction”, line 1, it reads: “Overall, most eyes had a final UDVA and CDVA that was higher than 80 EDTRS letters (Figure 4), while 78% of eyes achieved no change in CDVA (Figure 5). Finally, the postoperative refractive spherical equivalent of ±1.00 Dpt was achieved in 72% of eyes (Figure 6).”
Comment
The statements are confusing, because information is incomplete.
Consider modifying to: “Overall, most eyes in Groups 1 and 2 had a final UDVA and CDVA that was higher than 80 EDTRS letters (Include here the corresponding value in LogMAR) (Figure 4), while in these two groups no eye lost lines of CDVA and 22% gained one or more lines of CDVA (Figure 5). Finally, the
postoperative refractive spherical equivalent of ±1.00 Dpt was achieved in 72% of eyes
(Figure 6).”
Figure 4 legend. It reads: “Figure 4. Uncorrected distance visual acuity (UDVA) vs. corrected distance visual acuity (CDVA) in the 2 examined cumulative groups.”.
Comment: It should read: “Figure 4. Cumulative percentage of eyes with a given Uncorrected distance visual acuity (UDVA) vs. corrected distance visual acuity (CDVA), in all the cases included in Groups 1 and 2.”
In Figure 4, add the corresponding LogMAR values for each ETDRS letters visual acuity on the horizontal axis. Change the axis title to “Cumulative Visual Acuity (ETDRS letters / LogMAR).”
Figure 5 legend is too long.
Consider modifying to: “Figure 5. Change in corrected distance visual acuity (CDVA) in all the cases included in Groups 1 and 2.” The detailed explanation should be included in the text (see previous comment).
Figure 6 legend is too long. Consider modifying to: “Figure 6. Cumulative postoperative spherical equivalent refraction
in all the cases included in Groups 1 and 2.” All the other details should be mentioned in a paragraph in the “Results” section in the manuscript.
As mentioned earlier, lower-order aberrations, specifically Defocus and Astigmatism, are typically expressed in diopters rather than microns, as they can be easily determined clinically. These values are commonly used to calculate both the Spherical Equivalent and Defocus Equivalent. Therefore, I find the inclusion of the information in microns for 'Defocus' and 'Astigmatism' in Table 7 to be superfluous and potentially confusing for the reader. I recommend excluding these two parameters from the table to enhance visual clarity in the graphical representation. The information about “Defocus” and “Astigmatism” can be better interpreted simply analyzing Spherical Equivalent and Defocus Equivalent magnitudes in Diopters.
Table 2. Differentiation of UDVA, UCNVA, and TBUT parameters depending on the group.
As already mentioned, the Uncorrected Distance Visual Acuity (UDVA) in the control group, being entirely contingent upon refractive error, provides limited meaningful information and is recommended for deletion. Instead, a more relevant comparison could be established by including the preoperative Distance Corrected Visual Acuity of the control group. This would allow for an effective comparison with the postoperative UDVA of eyes implanted with the Lentis Mplus intraocular lenses.
Author Response
RESPONSE TO REVIEWER 2 COMMENTS
Comments and Suggestions for Authors
Comparative Analysis of the Visual, Refractive and Aberrometric Outcome with the Use of 2 Intraocular Refractive Segment Multifocal Lenses.
Thank you very much for taking the time to review this manuscript. Please find the detailed responses below and the corresponding revisions/corrections highlighted/in track changes in the re-submitted files.
COMMENT 1:
In page 1, in “Abstract”, line 5, it reads. “…were measured at scotopic pupil size in 42 eyes of patients implanted..”.
According to the description in “Examination” section, pharmacological mydriasis was used, therefore this is not a “scotopic pupil size”, which would be a naturally dilated pupil in low light conditions. This is a pharmacologically dilated pupil.
Therefore it should read: “…were measured through a pharmacologically dilated pupil in 42 eyes of patients implanted..”.
According to the descr
RESONSE 1: Thank you for pointing this out. We performed the measurements without pharmacologically induced mydriasis in a dark room environment. I pointed this out in the revised version of the manuscript.
COMMENT 2:
In page 1, in “Abstract”, line 7, it reads. “…and 107 eyes of control group. Results: No significant differences were noticed between the examined groups concerning UCDVA, UCNVA, and tear break-up time (p < 0.001).”
Comment:
Age-matched?
RESONSE 2: Thank you for pointing out this issue, we have included this information concerning the group age heterogeneity in the 12 paragraph of the discussion section.
COMMENT 3:
In page 1, in “Introduction”, first paragraph, first line, it reads: “Cataract surgery or refractive lens exchange with the implantation of multifocal intraocular lens (MF IOL) allows for good visual performance and spectacle independence [1]. The aim of the surgery is to provide rapid and complete visual rehabilitation with marginal postoperative refractive errors [2].”
Comment: Since the term “multifocal intraocular lenses” includes both bifocal and trifocal platforms, the statement is too broad and succinct. As several newer trifocal models have been marketed and successfully implanted in many patients, they should be at least mentioned and articles about their performance cited.
Therefore, consider modifying to: “Cataract surgery or refractive lens exchange with the implantation of multifocal intraocular lens (MF IOL) allows for good visual performance and spectacle independence, particularly for the newer trifocal platforms (AT Lisa tri 839 MP, Panoptix, Fine Vision, among others), which let the patients to have good unaided near, intermediate and far vision [1] (Breyer 2020; Niazi 2023; Galvis 2022; LubiÅ„ski2020; Ang 2023; Karam 2023). The aim of the surgery is to provide rapid and complete visual rehabilitation with marginal postoperative refractive errors, and good enough uncorrected visual acuity at the three distances. In cases of significant residual ametropia, additional photorefractive procedures may be needed [2].
New references to be cited:
Breyer DRH, Beckers L, Ax T, Kaymak H, Klabe K, Kretz FTA. Aktuelle Übersicht: multifokale Linsen und Extended-Depth-of-Focus-Intraokularlinsen [Current Review: Multifocal Intraocular Lenses and Extended Depth of Focus Intraocular Lenses]. Klin Monbl Augenheilkd. 2020;237(8):943-957. doi:10.1055/a-1111-9380
Niazi S, Gatzioufas Z, Dhubhghaill SN, et al. Association of Patient Satisfaction with Cataract Grading in Five Types of Multifocal IOLs [published online ahead of print, 2023 Oct 27]. Adv Ther. 2023;10.1007/s12325-023-02698-5. doi:10.1007/s12325-023-02698-5.
Galvis V, Escaf LC, Escaf LJ, et al. Visual and satisfaction results with implantation of the trifocal Panoptix® intraocular lens in cataract surgery. J Optom. 2022;15(3):219-227. doi:10.1016/j.optom.2021.05.002
Lubiński W, Podborączyńska-Jodko K, Kirkiewicz M, Mularczyk M, Post M. Comparison of visual outcomes after implantation of AtLisa tri 839 MP and Symfony intraocular lenses. Int Ophthalmol. 2020;40(10):2553-2562. doi:10.1007/s10792-020-01435-z
Ang RET. Long-term trifocal toric intraocular lens outcomes in Asian eyes after cataract surgery. J Cataract Refract Surg. 2023;49(8):832-839. doi:10.1097/j.jcrs.0000000000001195
Karam M, Alkhowaiter N, Alkhabbaz A, et al. Extended Depth of Focus Versus Trifocal for Intraocular Lens Implantation: An Updated Systematic Review and Meta-Analysis. Am J Ophthalmol. 2023;251:52-70. doi:10.1016/j.ajo.2023.01.024.
RESONSE 3: Thank you for pointing this issue. We agree that a more detailed explanation of different lens platforms could help the reader in this context.. We changed the manuscript accordingly and added some supporting references.
COMMENT 4:
In page 1, in “Introduction”, first paragraph, line 4, it reads: “Studies have reported excellent visual outcomes and low occurrence of photic phenomena following the use of multifocal rotational asymmetric intraocular lenses [3].”
Comment:
It is important that the reader has the concept of this different approach, rotational asymmetric intraocular lenses, and that unlike trifocal models, it does not yield good uncorrected intermediate vision.
Therefore, consider modifying to: “An additional approach has been the use of rotational asymmetric intraocular lenses. These platforms exhibit refractive rotational asymmetry unlike the standard bifocal or trifocal intraocular lenses, which are rotationally symmetric and based on the principles of diffraction and refraction. Studies have reported excellent visual outcomes and low occurrence of photic phenomena following the use of these multifocal rotational asymmetric intraocular lenses. The amount of addition is variable (usually +3.00 D for near vision, or +1.50 D for intermediate vision models). [3] (McNeely 2016; Oshika, 2029).”
Additional reference:
McNeely RN, Pazo E, Spence A, Richoz O, Nesbit MA, Moore TC, Moore JE. Comparison of the visual performance and quality of vision with combined symmetrical inferonasal near addition versus inferonasal and superotemporal placement of rotationally asymmetric refractive multifocal intraocular lenses. J Cataract Refract Surg. 2016 Dec;42(12):1721-1729. doi: 10.1016/j.jcrs.2016.10.016. PMID: 28007103.
Oshika T, Arai H, Fujita Y, Inamura M, Inoue Y, Noda T, Miyata K. One-year clinical evaluation of rotationally asymmetric multifocal intraocular lens with +1.5 diopters near addition. Sci Rep. 2019 Sep 11;9(1):13117. doi: 10.1038/s41598-019-49524-z. PMID: 31511557; PMCID: PMC6739307.
RESONSE 4: Thank you for pointing out this issue and for your kind suggestions. We have included this paragraph as suggested by the reviewer and provided the respective references in the revised version of the manuscript.
COMMENT 5:
In page 1, in “Introduction”, first paragraph, line 6, it reads: “Aberrations refer to the differences in the optical path-lengths (actual minus ideal wavefront achieved in the optical system). Optical aberrations of a specific visible light wavelength travelling through an optical system are generally classified into several categories: spherical refractive error (defocus), cylindrical refractive error, spherical aberration, coma, and other higher-order aberrations (HOA) [4]. Light scattering and HOA due to the refractive surface of the intraocular lens may lead to poor retinal image quality [5].”
Comment:
The statements are confusing and incomplete. Consider modifying to:
“Aberrations refer to differences in optical path lengths, which represent the deviation between the actual and ideal wavefront achieved in an optical system. Optical aberrations for a specific visible light wavelength traveling through an optical system are generally classified into two broad categories. The first category includes lower order aberrations, which account for most of the degradation in image quality. These lower order aberrations comprise spherical refractive error (defocus) and cylindrical refractive error. The second category is higher order aberrations (HOA), encompassing spherical aberration, coma, and others. When significantly high, HOA can impact the quality of vision and, notably, cannot be corrected with spectacles or contact lenses. [4]. Light scattering can also, along with HOAs, further degrade retinal image quality [5].”
RESONSE 5: We thank the reviewer for this advice and changed the manuscript accordingly
COMMENT 6
In page 1, in “Introduction”, first paragraph, line 12, it reads: “However, the optical aberrations may differ depending on the amount of near addition. The study aimed to compare visual, refractive, and aberrometric outcomes following surgery with the use of two refractive central-sparing rotational asymmetric multifocal lenses with near vision addition of +1.5 and +3.0 dpt.”
Comment.
The wording should be revised. In addition, it is important to mention that clinical objective measurements of IOLs performance (v.gr. aberrometry and double-pass optical visual quality analysis) have been questioned as to their real ability to evaluate the clinical performance of these platforms (Charman 2008; Fernández 2022; Vega 2023), and that they may be just a rough approximation of the in vivo performance of the IOLs as perceived by the patients. However, they can yield some useful information in some cases (D'Oria 2023). This issue should be mentioned both in “Introduction” and in more detail in “Discussion” section.
Therefore, consider modifying to:
“Optical aberrations may differ depending on many features of the intraocular lenses, including the amount of near addition. Although, currently there is not a consensus about the real ability of aberrometers to evaluate the clinical performance of multifocal intraocular lenses (Charman 2008; Fernández 2022; Vega 2023), it seems that they can provide useful information (D'Oria 2023). The present study aimed to compare visual, refractive, and aberrometric outcomes following surgery with the use of two refractive central-sparing rotational asymmetric multifocal intraocular lenses with two different amounts of near vision addition, specifically +1.5 versus +3.0 dpt.”
Additional References:
Charman WN, Montés-Micó R, Radhakrishnan H. Problems in the measurement of wavefront aberration for eyes implanted with diffractive bifocal and multifocal intraocular lenses. J Refract Surg. 2008 Mar;24(3):280-6. doi: 10.3928/1081597X-20080301-10. PMID: 18416263.
Fernández J, Rocha-de-Lossada C, Rodríguez-Vallejo M. Objective Optical Quality With Multifocal Intraocular Lenses Should Stop to Be Used or Cautiously Interpreted. Asia Pac J Ophthalmol (Phila). 2022;11(6):569. Published 2022 Nov 1. doi:10.1097/APO.0000000000000502.
Vega F, Faria-Ribeiro M, Armengol J, Millán MS. Pitfalls of Using NIR-Based Clinical Instruments to Test Eyes Implanted with Diffractive Intraocular Lenses. Diagnostics (Basel). 2023 Mar 27;13(7):1259. doi: 10.3390/diagnostics13071259. PMID: 37046477; PMCID: PMC10093131.
D'Oria F, Scotti G, Sborgia A, Boscia F, Alessio G. How Reliable Is Pyramidal Wavefront-Based Sensor Aberrometry in Measuring the In Vivo Optical Behaviour of Multifocal IOLs? Sensors (Basel). 2023 Mar 28;23(7):3534. doi: 10.3390/s23073534. PMID: 37050594; PMCID: PMC10099035.
RESONSE 6: Thank you for this comment and your recommendations. We included a respective paragraph which adds information with ref about the lack of consensus to aberrometrie’s ability to evaluate MFIOL performance.
COMMENT 7
Page 2, first paragraph following “Patients” subtitle, line 4. It reads: “The control group consisted of patients who were qualified for refractive surgery.”
Comment: It is very important to clarify here what type of refractive surgery (because indeed refractive lens exchange is a type of refractive surgery). I suppose the authors meant excimer laser corneal refractive surgery.
RESONSE 7: We agree with the reviewer and provided additional explanation about type of refractive surgery in the revised version of the manuscript.
COMMENT 8
Page 4, first paragraph, line 3. It reads: “In addition, uncorrected near visual acuity (UNVA) at 40 cm and best-corrected near visual acuity (BCNVA) at 40 cm were assessed using logMAR charts”
Comment:
Considering that models with two different additions were used (+1.5 versus +3.0 dpt), both near and intermediate uncorrected visual acuity should have been evaluated. If not, this weakness must be acknowledged.
RESONSE 8: Thank you for pointing this out. I agree that intermediate VA would have been of an important value. We have acknowledged it as a limitation in the discussion section. Thank you!
COMMENT 9
Page 4, second paragraph, line 1. It reads: “The iTrace measurements were performed after pharmacological mydriasis (scotopic pupil size) and a fixation target projected to infinity.”
Comment: A pharmacologically dilated pupil is not a “scotopic pupil”, which would be a naturally dilated pupil in darkness, without medication.
It is crucial to specify the mydriatic substance applied (such as phenylephrine, tropicamide, or cyclopentolate) and the corresponding doses used.
RESONSE 9: We agree with the reviewer and changed the respective paragraph accordingly (please see also comment …)
COMMENT 10
Another critical consideration is providing information on the diameter of the pupil used for aberrometric measurements. Aberrometry is linearly related to the measurement diameter, making this detail fundamental. If different diameters were employed for different eyes, it is essential to thoroughly review the data, establish a consistent diameter (commonly 6.00 mm), and then redo the analysis with the obtained values. Therefore in Table 1 it should be specified the diameter used to determine the aberrometric parameters.
RESONSE 10: We agree with the reviewer that wavefront parameters extracted from aberrometry always refer to a certain diameter of evaluation as all Zernike terms are ‘normalized’ to the unit circle. In this manuscript we referenced all Zernike terms of aberrometry to an optical zone diameter of 6 mm.
COMMENT 11
In Table 1, the head of the control group column, should be modified to “Control Group (preoperative measurement)”.
In Table 1, it is necessary to clarify whether “Coma” includes both, third order vertical and horizontal coma (Z3-1 and Z31) , whether “Trefoil” includes both, third order vertical and oblique trefoil (Z3-3 and Z33), and whether Spherical aberration includes only the Primary Spherical Aberration (Z40) or also the secondary spherical aberration (Z60).
Usually lower order aberrations, i.e. Defocus and Astigmatism, are not shown in microns, since they can be simply clinically determined in diopters, and used to calculate both Spherical Equivalent and Defocus equivalent. Therefore, I consider the information in microns is superfluous, and may be confusing for the reader.
The Uncorrected Distance Visual Acuity (UDVA) in the control group, being entirely contingent upon refractive error, provides limited meaningful information and is recommended for deletion. Instead, a more relevant comparison could be established by including the preoperative Distance Corrected Visual Acuity of the control group. This would allow for an effective comparison with the postoperative UDVA of eyes implanted with the Lentis Mplus intraocular lenses.
RESONSE 11: We thank the reviewer and changed the Table legend accordingly (we added the BCDVA in the table)
COMMENT 12
Page 5. Second paragraph following “Spherical equivalent refractive accuracy”, line 3. It reads:” The results of the analysis are presented by group in Table 1.”
Comment: This information should not be included in this section (“Methods”), but in the “Results” section.
RESONSE 12: We agree with the reviewer and moved this information to the Results section.
COMMENT 13
In addition, it would be interesting and useful for the reader to calculate and analyze the “Defocus Equivalent” of the eyes included in the study.
Holladay JT, Moran JR, Kezirian GM. Analysis of aggregate surgically induced refractive change, prediction error, and intraocular astigmatism. J Cataract Refract Surg. 2001 Jan;27(1):61-79. doi: 10.1016/s0886-3350(00)00796-3. PMID: 11165858.
RESONSE 13: We thank the reviewer for this suggestion. We have included the DEQ =sqrt(SEQ²+(0.5*cyl)²) in the table 1.
COMMENT 14
Page 6, first paragraph following “Visual Acuity and Refraction”, line 1, it reads: “Overall, most eyes had a final UDVA and CDVA that was higher than 80 EDTRS letters (Figure 4), while 78% of eyes achieved no change in CDVA (Figure 5). Finally, the postoperative refractive spherical equivalent of ±1.00 Dpt was achieved in 72% of eyes (Figure 6).”
Comment
The statements are confusing, because information is incomplete.
Consider modifying to: “Overall, most eyes in Groups 1 and 2 had a final UDVA and CDVA that was higher than 80 EDTRS letters (Include here the corresponding value in LogMAR) (Figure 4), while in these two groups no eye lost lines of CDVA and 22% gained one or more lines of CDVA (Figure 5). Finally, the
postoperative refractive spherical equivalent of ±1.00 Dpt was achieved in 72% of eyes
(Figure 6).”
RESPONSE 14: We thank the reviewer for this issue and changed this passages in the manuscript accordingly.
COMMENT 15
Figure 4 legend. It reads: “Figure 4. Uncorrected distance visual acuity (UDVA) vs. corrected distance visual acuity (CDVA) in the 2 examined cumulative groups.”.
Comment: It should read: “Figure 4. Cumulative percentage of eyes with a given Uncorrected distance visual acuity (UDVA) vs. corrected distance visual acuity (CDVA), in all the cases included in Groups 1 and 2.”
In Figure 4, add the corresponding LogMAR values for each ETDRS letters visual acuity on the horizontal axis. Change the axis title to “Cumulative Visual Acuity (ETDRS letters / LogMAR).”
RESPONSE 15: We thank the reviewer for this issue and changed this passages in the manuscript accordingly. We modified the text as suggested and added the logMAR units in addition to ETDRS letters.
COMMENT 16
Figure 5 legend is too long.
Consider modifying to: “Figure 5. Change in corrected distance visual acuity (CDVA) in all the cases included in Groups 1 and 2.” The detailed explanation should be included in the text (see previous comment).
RESPONSE 16: We thank the reviewer for this advice and shortened the legend accordingly. We made the modification and explanation in the text.
COMMENT 17
Figure 6 legend is too long. Consider modifying to: “Figure 6. Cumulative postoperative spherical equivalent refraction
in all the cases included in Groups 1 and 2.” All the other details should be mentioned in a paragraph in the “Results” section in the manuscript.
RESPONSE 17: We thank the reviewer for this advice and shortened the legend accordingly
COMMENT 18
As mentioned earlier, lower-order aberrations, specifically Defocus and Astigmatism, are typically expressed in diopters rather than microns, as they can be easily determined clinically. These values are commonly used to calculate both the Spherical Equivalent and Defocus Equivalent. Therefore, I find the inclusion of the information in microns for 'Defocus' and 'Astigmatism' in Table 7 to be superfluous and potentially confusing for the reader. I recommend excluding these two parameters from the table to enhance visual clarity in the graphical representation. The information about “Defocus” and “Astigmatism” can be better interpreted simply analyzing Spherical Equivalent and Defocus Equivalent magnitudes in Diopters.
RESPONSE 18: We followed the suggestions of the reviewer and excluded the Defocus and Astigmatism from the Table 7.
COMMENT 19
Table 2. Differentiation of UDVA, UCNVA, and TBUT parameters depending on the group.
As already mentioned, the Uncorrected Distance Visual Acuity (UDVA) in the control group, being entirely contingent upon refractive error, provides limited meaningful information and is recommended for deletion. Instead, a more relevant comparison could be established by including the preoperative Distance Corrected Visual Acuity of the control group. This would allow for an effective comparison with the postoperative UDVA of eyes implanted with the Lentis Mplus intraocular lenses.
RESPONSE 19: We followed the suggestions of the reviewer and added the preoperative BCDVA as reference in the table.
Reviewer 3 Report
Comments and Suggestions for Authors
Dear Authors
Thanks for the manuscript.
It would be beneficial in the discussion add a paragraph on the clinical recommendation. Between the 2 MF-IOL which cases would be more suitable for 1.5D and which for 3.0D?
Additionally, why scotopic measurements have been taken in pharmacological mydriasis and not physiological mydriasis?
Any differences between the groups in pre and post-op pupil diameters?
Author Response
Comments and Suggestions for Authors
Dear Authors
Thanks for the manuscript.
Thank you very much for taking the time to review this manuscript. Please find the detailed responses below and the corresponding revisions/corrections highlighted/in track changes in the re-submitted files.
COMMENT 1:
It would be beneficial in the discussion add a paragraph on the clinical recommendation. Between the 2 MF-IOL which cases would be more suitable for 1.5D and which for 3.0D?
RESPONSE 1: We thank the reviewer for this suggestion. We have added statement at the last paragraph of discussion section.
COMMENT 2:
Additionally, why scotopic measurements have been taken in pharmacological mydriasis and not physiological mydriasis?
RESPONSE 2: We thank the reviewer for pointing out this issue. We did not measure with pharmacological stimulated mydriasis but with physiological measurement in a dark room. We over-worked the respective paragraph in the manuscript.
COMMENT 3:
Any differences between the groups in pre and post-op pupil diameters?
RESPONSE 3: We thank the reviewer for this question. We have a post op pupil diameter for tested eyes. It showed differences in the groups. We added this data in the table 1 and mentioned the difference as a limitation. We did not measure with pharmacological stimulated mydriasis but with physiological measurement in a dark room. We over-worked the respective paragraph in the manuscript.
Reviewer 4 Report
Comments and Suggestions for Authors
Reviewer #: This is a very interesting article regarding the analysis of visual outcomes of 2 Intraocular Refractive Segment Multifocal Lenses. It is well known the expertise of the authors about this field and the paper is well written, however I feel that some improvements are necessary. For this reason I believe that a major revision is required before it can be accepted for publication.
1) Materials and Methods section, page 2: authors reported that informed consent was not required in their study. Why did they consider the consent unnecessary? Authors should carefully explain the reason to do not require a written consent, because informed consent is generally obtained in these type of studies.
2) Materials and Methods section, page 2: authors reported that “the inclusion criteria for the surgery were moderate or high refractive error not qualified for the laser corneal correction”. Authors should explain in details how they evaluate the eligibility for laser corneal correction (e.g. through corneal tomography or pachymetry analysis?).
3) Materials and Methods section, page 2: authors correctly excluded patients with previous keratorefractive surgery, but they should explain why. In fact keratorefractive surgery causes inaccurate measurement of anterior keratometry and the variation of keratometric index. therefore, it is necessary to use specific IOL power calculation methods (DOI: 10.3390/jcm12082890). I suggest explaining these reasons in the manuscript, with bibliography improvement.
4) Materials and Methods section, page 3: authors should specify which IOL Master model they used for the study: IOL Master 500 or IOL Master 700?
5) Materials and Methods section, page 3: authors used Hoffer-Q formula with cross check by Barrett Universal formula for eyes below the axial length of 22.00 mm for IOL power calculation. To be honest, Hoffer-Q formula should be preferred when axial length is below than 23.00mm (DOI: 10.1038/s41598-022-23665-0). Authors should explain the reason of their choice.
6) Materials and Methods section, page 3: I suppose that authors marked cornea at the slit lamp, they should report how did they made the horizontal marks.
7) Materials and Methods section, page 3: authors said that all patients with relevant posterior capsule opacification underwent ND:YAG laser capsulotomy. How many patients underwent YAG laser casulatomy? Of which groups?
8) Materials and methods section, page 4: authors performed best corrected visual acuity (BCDVA) at 5 m. Generally, the testing distance choice is 6m. Otherwise, they should correct the spherical equivalent.
9) Materials and Methods, page 5: Statistical analysis needs some improvement. In fact, preliminary required sample size is missing. This is a mistake. In addition, authors evaluate also bilateral eyes in their study. In case of bilateral eyes, smaller p-values could be obtained when they are evaluated in the same group. Authors used nested ANOVA test to overcome this problem, but the literature suggested other specific statistical methods in these cases, such as the mixed linear models, the Bootstrap or generalized estimating equations (DOI: 10.1016/j.ophtha.2020.05.017). Authors should add these concepts when discussing the limitations of the study, with bibliography improvement.
10) Results section, page 8: authors reported that “Concerning spherical equivalent refraction error, the group with the Lentis Mplus15 lens obtained significantly higher values than the control group (difference at the level of p = 0.016) and the postoperative group with the Lentis Mplus30 lens (difference at the level of p <0.001)”. Spherical equivalent error could be affected by IOL power calculation formula bias, authors should specify it in the text.
11) Figure 5: I fear that change in corrected distance visual acuity analysis could be affected by previous lens opacity. Please give more details about such analysis.
Comments on the Quality of English LanguageMinor editing of English language required
Author Response
RESPONSE TO REVIEWER 4 COMMENTS
Comments and Suggestions for Authors
Reviewer #: This is a very interesting article regarding the analysis of visual outcomes of 2 Intraocular Refractive Segment Multifocal Lenses. It is well known the expertise of the authors about this field and the paper is well written, however I feel that some improvements are necessary. For this reason I believe that a major revision is required before it can be accepted for publication.
Thank you very much for taking the time to review this manuscript. Please find the detailed responses below and the corresponding revisions/corrections highlighted/in track changes in the re-submitted files.
COMMENT 1:
1) Materials and Methods section, page 2: authors reported that informed consent was not required in their study. Why did they consider the consent unnecessary? Authors should carefully explain the reason to do not require a written consent, because informed consent is generally obtained in these type of studies.
RESPONSE 1: Thank you for pointing this out. The study was a retrospective analysis. According to the Declaration of Helsinki statements that if informed consent is impracticable to obtain the researchers can be done after approval of ethics committee. Some researchers also stated that the informed consent is not required to the retrospective studies because this kind of study meet the criteria that “the research involves no more than minimal risk to the subjects”. I have added this criteria in the text.
Block MI, Khitin LM, Sade RM. Ethical process in human research published in thoracic surgery journals. Ann Thorac Surg. 2006;82(1):6–12. doi: 10.1016/j.athoracsur.2006.01.084
COMMENT 2:
2) Materials and Methods section, page 2: authors reported that “the inclusion criteria for the surgery were moderate or high refractive error not qualified for the laser corneal correction”. Authors should explain in details how they evaluate the eligibility for laser corneal correction (e.g. through corneal tomography or pachymetry analysis?).
RESPONSE 2: We thank the reviewer for this advice. We have accordingly added in detail information in the material and methods section.
COMMENT 3:
3) Materials and Methods section, page 2: authors correctly excluded patients with previous keratorefractive surgery, but they should explain why. In fact keratorefractive surgery causes inaccurate measurement of anterior keratometry and the variation of keratometric index. therefore, it is necessary to use specific IOL power calculation methods (DOI: 10.3390/jcm12082890). I suggest explaining these reasons in the manuscript, with bibliography improvement.
RESPONSE 3: We agree with the reviewer and provided additional explanation and in detail information in the material and methods section along with the citation for this article.
COMMENT 4:
4) Materials and Methods section, page 3: authors should specify which IOL Master model they used for the study: IOL Master 500 or IOL Master 700?
RESPONSE 4: Thank you for this comment. We used IOL Master V.5.0 – it was a machine right before IOL Master 500 was released. It is highlighted in the text on second paragraph in section 2.2
COMMENT 5:
5) Materials and Methods section, page 3: authors used Hoffer-Q formula with cross check by Barrett Universal formula for eyes below the axial length of 22.00 mm for IOL power calculation. To be honest, Hoffer-Q formula should be preferred when axial length is below than 23.00mm (DOI: 10.1038/s41598-022-23665-0). Authors should explain the reason of their choice.
RESPONSE 5: Thank you for this comment. Our racionale is that in many publications the cutting point of formulas for short eyes is 22.00 (DOI: 10.1038/sj.eye.6702774, DOI: 10.1136/bjo.2008.148452, DOI: 10.1111/ceo.13058, DOI: https://doi.org/10.1038/sj.eye.6702774).
COMMENT 6:
6) Materials and Methods section, page 3: I suppose that authors marked cornea at the slit lamp, they should report how did they made the horizontal marks.
RESPONSE 6: Thank you for this comment. We have added additional information about how horizontal marking was made.
COMMENT 7:
7) Materials and Methods section, page 3: authors said that all patients with relevant posterior capsule opacification underwent ND:YAG laser capsulotomy. How many patients underwent YAG laser casulatomy? Of which groups?
RESPONSE 7: Thank you for this comment. We have added additional information about the numbers of patients and what we ment by relevant PCO.
COMMENT 8:
8) Materials and methods section, page 4: authors performed best corrected visual acuity (BCDVA) at 5 m. Generally, the testing distance choice is 6m. Otherwise, they should correct the spherical equivalent (DOI: 10.1097/j.jcrs.0000000000001057)
RESPONSE 8 Thank you for this comment. We have examined visual acuity at 5 m with the use of liquid cristal display testing charts with EDTRS letter size calibration to 5 m. We have added this information in the material and methods section.
COMMENT 9:
9) Materials and Methods, page 5: Statistical analysis needs some improvement. In fact, preliminary required sample size is missing. This is a mistake. In addition, authors evaluate also bilateral eyes in their study. In case of bilateral eyes, smaller p-values could be obtained when they are evaluated in the same group. Authors used nested ANOVA test to overcome this problem, but the literature suggested other specific statistical methods in these cases, such as the mixed linear models, the Bootstrap or generalized estimating equations (DOI: 10.1016/j.ophtha.2020.05.017). Authors should add these concepts when discussing the limitations of the study, with bibliography improvement.
RESPONSE 9 Thank you kindly for this comment. We acknowledged this important limitation with bibliography improvement.
COMMENT 10:
10) Results section, page 8: authors reported that “Concerning spherical equivalent refraction error, the group with the Lentis Mplus15 lens obtained significantly higher values than the control group (difference at the level of p = 0.016) and the postoperative group with the Lentis Mplus30 lens (difference at the level of p <0.001)”. Spherical equivalent error could be affected by IOL power calculation formula bias, authors should specify it in the text.
RESPONSE 10: We thank the reviewer for this advice. We have added additional specification in the manuscript in the discussion section.
COMMENT 11
11) Figure 5: I fear that change in corrected distance visual acuity analysis could be affected by previous lens opacity. Please give more details about such analysis.
RESPONSE 11: Thank you for this comment. We agree about the probable change in CDVA being also affected by previous lens opacity. As described in Material and Methods section some patient enrolled in to the study had incipient or moderate cataract.
Round 2
Reviewer 2 Report
Comments and Suggestions for Authors
Most of the adjustments were performed
Author Response
Dear Reviewer. Thank you for your important input and time to mark your comments. With kind regards. Markuszewski
Reviewer 4 Report
Comments and Suggestions for Authors#Reviewer: Authors addressed almost all my requests and now the manuscript is considerably improved. There is only a little mistake regarding the bibliography.
In fact, at the endo the manuscript they correctly reported that in case of bilateral eyes there are other specific statistical methods such as mixed linear models, the Bootstrap or generalized estimating equations that could be used for statistical analysis, but the supporting reference of these concepts [Reference N°42] is wrong. In fact, the paper with PMID: 31561878 does not talk about correct statistic tests in case of bilateral eyes, but the article with PMID: 32739185 does.
Author Response
Dear Reviewer. Thank you for your comment. I have changed the reference accordingly. Thank you.